# OpenReview forum: "From Curiosity to Caution: Mitigating Reward Hacking for Best-of-$N$ with Pessimism"
_ICLR.cc/2026/Conference — ICLR 2026 Poster_

### Official Review · Reviewer_nUU2 · 2025-10-15

**Soundness:** 2
**Presentation:** 3
**Contribution:** 2
**Rating:** 4
**Confidence:** 4

**Summary:**

The paper proposes Best-of-N with pessimism to mitigate potential reward hacking issues in test-time scaling. Specifically, they capture "caution" by training a lightweight predictor network to reconstruct reward model features, and the pessimistic score is larger when the prediction error is lower (the reward model is more familiar with the response pattern). Experiment results on GSM8K, Math-500 and BBH-hard demonstrate that their methods are stronger than the Best-of-N baseline.

**Strengths:**

- The proposed pessimistic score is reasonable, which accounts for the reward model's familiarity with the input. The reward design thus becomes the original reward minus the prediction error, and is clear and easy to implement.
- Experiment results demonstrate that the proposed pessimistic score is effective when applied to Best-of-N method. What's more, it is also useful for OOD domain test-time scaling.

**Weaknesses:**

- The authors did not compare with other strong test-time scaling methods, such as self-consistency. So it is unsure whether their method is widely applicable to other test-time scaling methods, as well as state-of-the-art inference time scaling methods.
- The proposed method is not as lightweight as the authors claimed. In fact, it requires extra training for each reward model. That means for every reward model on a specific task, they need to train a predictor network again based on the specific reward model.
- The layer number L is a hyperparameter, and it is unclear how they select it. The authors only mentioned in appendix that they use L=10. It is unsure whether this hyperparameter can be applied to other settings / domains / reward models.

**Questions:**

See weaknesses above.

---

> ### Author Response · Authors · 2025-11-22
> **Rebuttal to nUU2**
>
> Thank you for your thoughtful review and recognition of our approach's clarity and effectiveness. We'll address each of your concerns below.
>
> **W1: Comparison with other test-time scaling methods**: We want to clarify that **our work specifically focuses on reward-model-guided Best-of-N sampling, which is a distinct paradigm from methods like self-consistency**. Self-consistency aggregates multiple reasoning paths through majority voting without using reward models at all, while BoN uses learned reward models to score and select responses. BoN is also a popular paradigm for RLHF, especially for algorithms like GRPO where it's like optimizing towards a variant of BoN sampling. However, we do recognize the necessity to include more comparison with more baselines similar to our setting, like χ²-regularization, soft BoN, etc. We're now adding explicit comparison curves in our figures to make this clearer in the revised version, and these results in Figure 8 of Appendix G in our revised manuscript show that our pessimism consistently outperforms new baselines.
>
> **W2: Training overhead and practicality**: This is a fair concern that deserves clarification. While we do need to train a predictor network for each reward model, the training requirements are actually quite modest compared to reward model training itself. Also, our results in Figure 3 indicate that when facing tasks with similar prompting patterns, the pessimism weights does show some generalization. Moreover, in similar settings like reward model guided BoN sampling, or even RL training workflows, the compute bottleneck is actually generating the N responses (or rollouts) as these require auto-regressive generation, and reward models and our pessimism typically require only a single forward pass and this is not really the bottleneck. Table 4 in the appendix contains all training related setup, including VRAM usage, our experiments related to training could be done with a minimum of 16GB VRAM but ideally require 24GB to prevent OOM when facing longer responses.
>
> **W3: hyperparameter selection**. Thank you for your concern on hyperparameter selection, we have now included experiments comparing different layers, in the Appendix E of our revised version.

---

> > ### Comment · Reviewer_nUU2 · 2025-11-24
> >
> > Thanks for your rebuttal. However, I am not convinced and will retain my original score.
> >
> > W1: I obviously know that self-consistency and Best-of-N are distinct methods, and self-consistency does not require a reward model as Best-of-N does. But Self-Consistency is a widely used method in test-time scaling, and as self-consistency series of works do not need to load an additional reward model, this makes them more convenient and cheaper to use then Best-of-N. As long as the authors are improving techniques under the big topic of test-time scaling, they should not avoid comparing with such commonly seen and foundational baselines.
> >
> > W3: The authors' ablation study in Table 6 is not rigorous. They only use a subset of GSM8K (100 questions) and it is obvious that this subset cannot fully reflect the full test set. Their method's performance in Table 6 is 84 while the same method's performance in Table 1 (main result) on the whole test set is 81.1+/-0.1. This clearly shows that their ablation study in the subset of Table 6 is not representative for the whole test set. What's more, based on their ablation study results, their method is very sensitive to the choice of the layer L (performance ranges from 77 to 84).

---

> > > ### Author Response · Authors · 2025-11-25
> > > **Response to nUU2**
> > >
> > > Thank you for your continued engagement and for raising these important points. We appreciate the opportunity to clarify our reasoning.
> > >
> > > W1: Comparison with self-consistency: We respectfully maintain that self-consistency falls outside the scope of our work which is to mitigate *reward hacking*, and this is not an arbitrary choice but rather follows established conventions in the reward hacking literature. Specifically, the relevant works in mitigating reward hacking we compare against as requested by other reviewers [1,2,3] do not include self-consistency as a baseline either. The reason is fundamental: **self-consistency does not involve reward models at all**, making it orthogonal to reward hacking mitigation. Our contribution specifically addresses the failure modes of reward-guided selection, while self-consistency is primarily a prompting and voting technique that's not relevant to rewards.
> > >
> > > Beyond scope considerations, there are practical reasons why self-consistency is not directly comparable. First, *self-consistency is inherently task-specific*: it requires specialized decoding logic to replace the greedy decode in CoT prompting by sampling from the language model’s decoder to generate reasoning paths separate from answers, then parse and generate sets of answers. In contrast, reward-based methods provide a unified framework across domains without task-specific engineering. For certain tasks where answers are not discrete, or involves longer spans of text, the majority voting on answers may no longer work. Second, *self-consistency cannot operate in online settings* where responses must be selected given the full response set, since majority voting requires collecting all candidates before making a decision. Reward-based selection, including our approach, naturally supports online deployment like accepting any response with rewards higher than certain values. We believe these distinctions justify our focus on reward-based baselines.
> > >
> > > W3: Ablation study representativeness: You raise a valid concern about the subset size. We acknowledge that the 100-question (with 100 responses per problem) subset yields different absolute accuracy values compared to the full test set. However, we want to emphasize that **the relative trends remain consistent and informative**. As you noted, the performance range across layer choices indicates sensitivity to L, but this sensitivity follows a clear pattern: *deeper layers consistently yield better results*. This trend, rather than the absolute numbers, is what the ablation is designed to reveal. Given the computational constraints gathering 10000 responses along with rewards within the rebuttal timeframe, we prioritized demonstrating the existence and direction of this trend over achieving perfect alignment with full-dataset numbers. We will clarify this limitation in the revised manuscript and, resources permitting, expand the ablation to larger subsets for the camera-ready version.
> > >
> > >
> > > [1] Inference-Time Reward Hacking in Large Language Models (NeurIPS'25)
> > > [2] Is Best-of-N the Best of Them? Coverage, Scaling, and Optimality in Inference-Time Alignment (ICML'25)
> > > [3] Soft Best-of-n Sampling for Model Alignment (ISIT'25)

---

### Official Review · Reviewer_JSWD · 2025-10-25

**Soundness:** 4
**Presentation:** 4
**Contribution:** 3
**Rating:** 8
**Confidence:** 4

**Summary:**

This paper addresses reward hacking in Best-of-N (BoN) sampling. The authors introduce an elegant, lightweight, and practical solution called "caution."

The method works by training a lightweight predictor network to mimic the internal features of a given reward model on in-distribution data. At inference time, the predictor's error on a new candidate response serves as an uncertainty penalty, which is subtracted from the original reward score. Extensive experiments demonstrate that this approach effectively mitigates reward hacking, leading to improved performance with N.

**Strengths:**

- A core strength is the paper's excellent presentation and framing of the problem. Drawing a parallel between curiosity-driven exploration in RL and "caution" for avoiding OOD reward exploitation is an insightful and generally useful contribution.
- The proposed solution is compelling in its simplicity and practicality. Training the auxiliary predictor network is a one-time offline cost (with unlabeled data), and the inference-time overhead is minimal as it leverages the same internal representations as the reward model.
- The experiments are extensive and well-designed.

**Weaknesses:**

W1. This is both a perceived weakness and a question. Please correct me if I misunderstood the setup. The "caution" mechanism defines "in-distribution" by learning the typical feature patterns of the base reward model. This is effective at penalizing novel responses that exploit weird loopholes, but it may fail to mitigate, and could even reinforce, the reward model's own systemic biases which you mention in Lines 81-83 and in Figure 4. For instance, if a reward model was trained on data where humans consistently preferred verbose answers, its internal features would encode verbosity as a "normal" characteristic of high-quality responses. The caution predictor would learn this pattern and would consequently fail to penalize a verbose, incorrect, reward-hacking response. It might even penalize a concise, correct response for being stylistically "out-of-distribution." In essence, the method regularizes generations toward the central tendencies of the reward model, and if those tendencies are themselves flawed, the method risks preserving those flaws. If doing so is indeed shown to mitigate hacking, why do you think hacking happens to begin with?

W2. What is the most fundamental way to combine a reward signal and an uncertainty signal with BoN? Subtracting the uncertainty from the reward feels very natural from an RL / training point of view. However, BoN only cares about the ranking of the responses according to their rewards, making it insensitive to any shifts or rescaling. Is there a way we can introduce uncertainty in the ranking itself and how would you justify subtraction for inference-time alignment? I think this aspect is worth discussing in the paper and makes your decision of running a z-score transformation followed by using a global hyperparameter to balance the subtraction of the reward with the uncertainty more convincing.

Other work considered, for example, a multiplicative exponential term to model uncertainty in reward scoring and they motivate it well for policy optimization [1].

[1] Houliston, Sam, et al. "Uncertainty-penalized direct preference optimization." arXiv preprint arXiv:2410.20187 (2024).

**Questions:**

- How do you expect the "caution" mechanism to perform if the base reward model is systematically biased?

- The paper's discussion of Huang et al. (2025b) is very clear. A direct empirical comparison on the same experimental setup would be appreciated, as it represents the most relevant alternative approach.

- Why do you not consider calibrated reward as done in InfAlign (Balashankar et al., 2024)? Since Best of n only cares about the ranking of the reward scores, such a calibration is theoretically motivated. Instead, you opt for a z-score transformation. Can you elaborate on lines 300-302? In particular, did you perform the z-score transformation per prompt or across prompts?

- You may consider citing [2]. They use the reward model's learned representation of the prompt response pair to model its uncertainty. They argue when a data point is (almost) in distribution, there is less uncertainty.

[2] Zhang, X., Ton, J. F., Shen, W., Wang, H., & Liu, Y. (2024). Overcoming reward overoptimization via adversarial policy optimization with lightweight uncertainty estimation. arXiv preprint arXiv:2403.05171.

---

> ### Author Response · Authors · 2025-11-22
> **Rebuttal to JSWD**
>
> Thank you for your thoughtful review and for recognizing our work's clarity and contributions. We'll address each of your points below.
>
> **W1/Q1: Caution mechanism and systematic biases:** You raise an important concern, but there's a key distinction we'd like to clarify. Our method penalizes responses that are distributionally atypical, not those that deviate from the reward model's preferences. The crucial insight is that reward hacking responses are themselves out-of-distribution, they exploit spurious correlations in ways that go beyond typical training patterns.
>
> Consider your verbosity example. Were a reward model trained on verbose responses, then normal in-distribution responses would indeed be verbose. But reward hacking doesn't occur through simple verbosity, it happens when responses become *atypically* verbose or exploit patterns in unusual ways, for instance, generating a lot of hallucinative spans but benefit from length bias. Figure 4 demonstrates this: the verbose, incorrect response that scores highly (\#1/1000) in reward scores actually receives a high pessimism penalty because it fails to follow standard formatting requirements.
>
> The fundamental reason reward hacking happens is that as N increases, BoN drifts into OOD regions where the reward model becomes unreliable. Our method identifies and penalizes exactly these OOD regions, keeping selection where the reward model's correlations remain valid.
>
> **W2: Justification for subtraction:** Excellent point about BoN's invariance to affine transformations. Inspired by RL methods, we decided to use subtraction and use a hyperparameter to impact the reward signal. We agree that there's definitely more room of discussion on which strategy combines the reward signal with a OOD indicator, and that might be an interesting point for future work.
>
> **Q2: Direct comparison with Huang et al. (2025b):** We're now adding explicit comparison curves with their χ²-regularization approach in Figure 8 of Appendix G in our revised manuscript. Our results show that pessimism achieves better peak accuracy and less degradation at large N because we leverage beyond-worst-case structure in how reward models fail.
>
> **Q3: Z-score normalization and calibrated rewards:** We thank the reviewer for pointing out the calibrated reward approach from InfAlign (Balashankar et al., 2024). We will discuss and cite this work in our revised manuscript. We opted for z-score normalization for its simplicity and computational efficiency, while still ensuring that reward model scores and RND scores are on comparable scales when combining them. Since Best-of-N selection depends only on relative rankings within each prompt, both calibration approaches preserve the necessary ranking structure.
>
> Regarding lines 300-302, we performed z-score transformation across prompts (i.e., using global statistics computed over all responses from all problems in the dataset). We conducted an ablation study comparing per-prompt versus across-prompts normalization and found that while per-prompt normalization leads to minor differences in absolute accuracy values (within 0.5-2.0 percentage points), our main conclusions about the effectiveness of RND-based regularization and χ²-based pessimism remain unchanged under both normalization schemes.
>
> **Q4: Zhang et al. (2024):** Thank you for this relevant reference. We're adding this citation and discussion in our revised manuscript as it indeed shares the key insight that reward model representations encode useful uncertainty information, which we extend through the curiosity/RND framework for inference-time reward hacking mitigation.

---

> > ### Comment · Reviewer_JSWD · 2025-11-27
> >
> > W1/Q1: This makes a lot of sense, thank you for the clarification
> > Q3: Thanks for this experiment
> >
> > I strongly support the acceptance of your paper! I thank the authors for their good work.

---

### Official Review · Reviewer_7MEf · 2025-10-27

**Soundness:** 4
**Presentation:** 4
**Contribution:** 4
**Rating:** 8
**Confidence:** 3

**Summary:**

In this work, the authors mitigate reward hacking by adding a pessimism or caution term to a reward model. Specifically, they focus on Best-of-N sampling and mitigate the selection of high proxy-reward, low true-reward generations by penalizing out-of-distribution samples where the proxy-reward estimate is uncertain and inaccurate. They do so by training a second network to reconstruct the proxy-reward model's latent representation. Then, on future data, they use the error of this reconstruction as a measure of the certainty of the reward model. They regularize their reward with this certainty term, penalizing OOD samples which result in low reconstruction while having abnormally high reward.

**Strengths:**

* The method is intuitive and interesting in its own right: quantifying OOD-ness of data using a second model trained with reconstruction loss
 * The paper is clear and well-written and provides a thorough study of their method and demonstrates its efficacy.
 * The empirical results demonstrate hacking mitigation on mathematical reasoning tasks.

**Weaknesses:**

* The authors say they see "monotonic" performance improvements for all N, yet in my opinion the curves look more like a plateauing reward. Indeed, past work on hacking often frames the approach as avoiding over-optimization and recovering an "optimal" reward rather than expecting gains as N increases forever [1]. I don't think this claim is necessary for the impact of the paper, and I think some may disagree with it.
 * Notationally, it was a little confusing introducing $T$ as separate from $r$. I understand that you take some intermediate layer from $r$, but I might define that more explicitly in the paragraph on line 202 or streamline notation.

[1] Khalaf, H., Verdun, C. M., Oesterling, A., Lakkaraju, H., & Calmon, F. D. P. (2025). Inference-Time Reward Hacking in Large Language Models

**Questions:**

* In Table 2, it seems you use $\lambda$ as a convex combination of reward terms, $\lambda r(x) + (1-\lambda) u(x)$. This is different than your regularization term introduced in line 236. Can you standardize this?
 * Intuitively why do we expect any performance gains when using caution only? We aren't using any reward signal at all, and are just picking completions that are more in distribution. What are the costs in terms of coherency or diversity in completion?
 * In Table 3, for Lightweight + Separate Emb.: How can Peak Acc be lower than Final Acc?
 * In your proofs, you condition on $\theta$, but isn't $\theta$ not a random variable? Why are you conditioning on it?
 * Can you explain in Prop. 2 where you use the $E[r^*|\hat{r}]$?

---

> ### Author Response · Authors · 2025-11-22
> **Rebuttal to 7MEf**
>
> Thank you for your thoughtful review and recognition of our work's clarity and contributions. We appreciate your constructive feedback and will address each of your points below.
>
> **W1: Claims of "monotonic" performance improvements**: Thank you for pointing this out, these curves don't show strictly monotonic improvement, and we acknowledge that a more precise description would be: substantially reduced degradation compared to standard reward-model-only selection, rather than perfect monotonicity across all N. This is expected behavior since reward models are inherently imperfect. We will revise the manuscript to more accurately describe this as "maintaining stable or improving performance across most instances" rather than claiming strict monotonicity.
>
> **W2 / Q1: Notation clarity for T, Standardization of λ :** Thank you for catching this notational inconsistency. We're incorporating notation improvements in our revised version.
>
> **Q2: Performance gains with caution only and potential costs:** Your intuition is insightful. Our approach can be understood as performing distributional selection analogous to natural selection: our pessimism favor responses that are better "adapted" to the pessimism's training environment by penalizing distributional deviation. When using caution only (λ=1.0), we're essentially selecting responses most in-distribution from the reward model's perspective without any reward guidance.
>
> This can still yield improvements, particularly when facing extremely difficult tasks that are out-of-distribution or out-of-domain for the reward model. In these scenarios, the reward model itself often makes negative contributions as N scales up, essentially selecting responses according to its own training preference distribution (which emphasizes superficial patterns like verbosity or formatting styles) rather than true accuracy. Here, pessimism effectively acts as soft prompting, as it naturally filters out responses that don't follow basic formatting requirements and task specifications, which tend to be OOD from the reward model's perspective: the reward model is unreliable on out-of-domain task, so its signal is harmful rather than helpful. However, when the reward model is reasonably reliable (as on GSM8K), the combined approach (moderate λ ≈ 0.6-0.8) performs best by balancing reward maximization with distributional constraint, we get the benefits of the reward signal while still filtering out exploitation of spurious correlations.
>
> **Q3: Peak Acc vs Final Acc in Table 3:** Thank you for catching what appears to be a confusing presentation. This is due to a typo and the correct value would be swapping these two. For all our visualizations we use scripts from a fixed data-source, while for this specific table we draw manually.
>
> **Q4: Conditioning on $\\theta$**
>
> Thank you for pointing this out.  Note that the main result that puts all of the pieces together, Theorem 3, does not condition on $\\theta$, which is indeed random.  Conditioning on $\\theta$ provides stronger guarantees that can elucidate the role that OOD behavior in the data have on the necessity of pessimism; indeed, in Proposition 2, it is clear from the final expression that the greedy (BoN) selection approach is only bad when there is a substantial portion of  the norm $\\hat \\theta$ comes from the orthogonal complement of the known distribution.  We will add discussion in the camera ready draft to help clarify this point, but please let us know if it is still unclear.
>
> **Q5: Proof of Proposition 2**
>
> We apologize for the lack of clarity.  The proof proceeds by first computing $\\mathbb E\[r^\* | \\hat r\]$, then explicitly computing $\\mathbb E\[\\hat r | \\theta^\*, \\hat \\theta\]$ using the Gaussianity assumption.  We then use Proposition 1 to compute $\\mathbb E\[r^\* | \\theta^\*, \\hat \\theta\]$ and use the linearity of expectation to compute $\\mathbb E\[r^\* \- \\hat r | \\theta^\*, \\hat \\theta\]$.  We apologize for the lack of clarity and will add in more description in the camera ready.

---

> > ### Comment · Reviewer_7MEf · 2025-11-24
> >
> > Thank you for your responses! I appreciate the work the authors have put into the paper and their response, and I believe this paper deserves to be accepted to ICLR.

---

### Official Review · Reviewer_sWFV · 2025-10-31

**Soundness:** 2
**Presentation:** 3
**Contribution:** 3
**Rating:** 4
**Confidence:** 3

**Summary:**

* The paper aims to improve the degradation in performance in the Best-of-N sampling scheme for large values of $N$ ("reward hacking") by constructing a weighted sampling scheme which combines reward and uncertainty scores.
* In the model, a language model stochastically maps a prompt $x$ to a response $y$, a reward model $r$ is a mapping from prompt-response pairs $(x,y)$ to scores. Given reward model $r$, uncertainty estimate $u$ and hyper-parameter $\\lambda$, the proposed approach samples candidates according to the “pessimistic” reward $r_\\mathrm{LCB}=r-\\lambda u$. The uncertainty estimate is inspired by random network distillation, aiming to detect OOD datapoints.
* A theoretical analysis is briefly described in the body of the paper, and presents a simplified setting in which pessimistic sampling converges towards the optimal-reward response but the absolute gap between naive and pessimistic sampling grows with $N$.
* The empirical analysis evaluates performance on three datasets, showing favorable performance compared to the Best-of-$N$ baseline.

**Strengths:**

* Addresses a well-motivated and timely issue.
* Proposed method seems relatively simple to implement, suggesting potential practical applicability.
* The presentation is generally clear and easy to follow.

**Weaknesses:**

* Despite the focus on empirical analysis, the analysis code doesn’t seem to be attached.
* While related prior work is mentioned (e.g., Huang et al. 2025, Jinaai et al. 2024), empirical evaluation doesn’t seem to include it as a possible baseline.
* The theoretical analysis underlying section 2.3 seems to be relatively intricate, but the body of the paper presents it only briefly. In particular, key assumptions, their limitations, implications of the analysis, and proof techniques are do not seem to be discussed.

**Questions:**

* Is it possible to elaborate on the intuitive assumptions and limitations underlying the theoretical analysis?
* What are the general limitations of the proposed approach, and when is it expected to fail?
* How does the method handle cases in which the ground truth reward has “true" label noise? (e.g., topic in which annotators have typically diverse preferences).
* Is it possible to formulate the relation the proposed method as a "soft distance constraint" between the training set and candidate response distributions? For example, if I understand correctly, it seems that only "in-distribution" candidates will be sampled when $\\lambda \\to \\infty$ and $u$ is a perfect OOD detector, because any OOD candidates will be penalized heavily. Does this intuition hold? And if so, what can be said about intermediate values of $\\lambda$?

---

> ### Author Response · Authors · 2025-11-22
> **Rebuttal to sWFV**
>
> Thank you for your thoughtful review and constructive feedback. We'll address each of your concerns and questions below.
>
> **W1: Code availability**
>
> Thank you for pointing this out. We have now anonymously released the full codebase, including scripts and step-by-step guide to reproduce our main results: [https://anonymous.4open.science/r/from-curiosity-to-caution-iclr/](https://anonymous.4open.science/r/from-curiosity-to-caution-iclr/)
>
> **W2: Direct comparison with related works**
>
> This is a good point; we have added a direct comparison between our method and baselines like χ²-regularization and soft BoN. Our results show that our pessimism consistently outperforms new baselines, with details in Figure 8 of Appendix G in our revised manuscript.
>
> **W3: Lack of proof techniques discussed in the main text**
>
> Thank you for pointing this out; we defer the discussion of the proof to the appendix due to space constraints, although we will add a high-level discussion in the camera ready when given an additional page.  For a summary of that discussion, please see Q1.
>
> **Q1: Assumptions, limitations of theoretical analysis**
>
> The theoretical analysis is not intended to represent a fully realistic setting but is rather there to help illustrate the general principle of caution in a simple, analytically tractable regime.  As a result, in our theory we consider only two-layer MLPs as our teachers and students with only the first layer trainable.  We also make a strong assumption on the data, that it is supported on a low dimensional linear subspace and thus out-of-distribution detection amounts to measuring the norm of a vector in the orthogonal complement. The key result says that our approach successfully achieves this (up to constants) with high probability when trained to convergence and that when applied to an offline linear bandit problem, our approach can attain optimal expected reward.
>
> **Q2/3: General limitations and failure mode**
>
> In practice, our method is effective when there exists a notion of "typical" or "in-distribution" responses that the reward model handles well, and reward hacking arises from OOD responses that exploit spurious correlations which we do observe in our experiments. However, due to the imperfect nature of RMs themselves, when such OOD measurement no longer correlates to hacking, our method would generally degrade. This also includes cases where the ground truths are not actually preference oracles, and we cannot generally "guess" the diverse preference behind each label.
>
> Q4: Your intuition is largely correct and insightful. **When λ is very large and detector perfectly detects OOD samples, our method would indeed restrict sampling to essentially in-distribution candidates** . However we have to note that the optimal choice of λ does rely on RMs and tasks,; the empirical results in Table 2 indicate that a larger lambda (like 0.8) does exhibit better performance, when the reward model is less reliable.

---

> > ### Comment · Reviewer_sWFV · 2025-11-27
> >
> > Thank you for the detailed response! The rebuttal addressed my main concerns, and I am therefore increasing my original score.

---

### Official Review · Reviewer_4Xb3 · 2025-11-01

**Soundness:** 3
**Presentation:** 4
**Contribution:** 3
**Rating:** 6
**Confidence:** 3

**Summary:**

The paper addresses the problem of reward hacking in Best-of-N sampling, where due to biases of the reward model sub optimal preferences might be made. The authors propose "caution," an inference-time technique that instantiates the principle of pessimism from reinforcement learning. It works by training a lightweight "predictor" network to predict the internal feature representations of the frozen reward model on a dataset of typical responses. The prediction error of this network is then used as an uncertainty score. This score is subtracted from the reward model's output to create a "pessimistic" reward, which penalizes responses that are out-of-distribution (OOD) from the perspective of the reward model. The authors demonstrate this approach with significant gains over GSM8K, MATH-500, BigBench-Hard reasoning tasks.

**Strengths:**

- The paper addresses an important and persistent problem in reward hacking that exists despite a large amount of papers on the subject.
- The experimental evaluation is comprehensive and well-designed. The testing across different distributions and domains (GSM8K, MATH-500, BBH) convincingly demonstrates the robustness of the approach.
- The ablation studies in Section 3.2 are thorough and successfully validate the key design decisions, especially the critical importance of using reward model features over random ones.
- The proposed method is computationally efficient and practical. The predictor network is trained fully offline, and at inference time, its forward pass can be parallelized with the reward model's, adding minimal overhead. This makes it a readily applicable solution for practitioners already using BoN sampling.

**Weaknesses:**

- Weak theoretical justification, It is unclear how the insights from this linear-Gaussian model translate to the complex, high-dimensional geometry of transformer feature spaces.
- Stronger baselines. The paper could be strengthened by comparing "caution" to other plausible inference-time uncertainty estimation or OOD detection techniques. There has also been a lot of work in reward hacking mitigation, so a comparison with more techniques would be appreciated.
- Qualitative examples - the paper would benefit from more qualitative study of the hacking mitigations
- Human alignment - more human evaluation would strengthen the usefulness of the approach for most tasks.

**Questions:**

- The main comparison is to standard BoN. How might "caution" compare to other classes of OOD/uncertainty detection methods applied at inference time, such as using Monte Carlo dropout on the RM's final layers or calculating Mahalanobis distance in the feature space of the RM to penalize outliers?
- The predictor Pθ is trained on typical responses from one policy (Llama-3.2-3B). How well would this pre-trained predictor generalize if used to score responses from a different, more capable base model?
- he case study highlights the detection of formatting errors. Can you provide examples of more semantic or stylistic deviations that the method successfully identifies and penalizes?

---

> ### Author Response · Authors · 2025-11-22
> **Rebuttal to 4Xb3**
>
> Thank you for your thoughtful review and constructive feedback. We appreciate your recognition of our contributions and will address each of your concerns below.
>
> **W1: Weak theoretical justification:** We certainly agree that the linear-Gaussian model and two-layer MLPs do not suffice to describe practical scenarios; our theoretical justification is meant more as an illustration in an analytically tractable setting that our approach is reasonable, wherein we show that in this simple setting, our method achieves strong OOD detection and performance in an offline linear bandit problem.
>
> **W2/Q1: Stronger baselines:** Thank you for this excellent suggestion. We have now included direct comparisons with χ²-regularization and soft BoN approaches as you recommended. **These new comparison curves in Figure 8 of Appendix G in our revised manuscript show that our pessimism consistently outperforms these baselines**, achieving better peak accuracy and substantially reduced degradation at large N. We believe these additional comparisons strengthen our empirical case considerably and have added them to the revised manuscript.
>
> **W3/Q3: Qualitative examples and semantic deviations:** You make an important point about the need for diverse qualitative analysis. Beyond the formatting examples we highlighted in our main text, we actually provide more diverse examples along with visualized distribution of reward / pessimism scores that is considered as hacking / normal responses, in Figure 5 in our Appendix. Although normal responses that follow the guidelines as prompted, hacking examples do exhibit significantly more diverse, OOD patterns that aren't easily identified, and this is the exact reason why our work is necessary, as we attempt to mitigate hacking by penalizing OOD samples. Identifying and qualitatively analyzing reward hacking patterns could be a very interesting direction for future work, but it might be highly task-specific.
>
> **W4: Human alignment and evaluation:** This is an excellent point. While our current evaluation focuses on reasoning tasks with where human evaluation is less critical, we completely agree that for more subjective tasks like creative writing or open-ended dialogue, human evaluation would be essential to validate that our method truly improves alignment rather than just optimizing proxy metrics.
>
> **Q2: Generalization across base models:** Great question. Our approach is essentially performing distributional selection, analogous to natural selection favoring responses that are better adapted to the reward model's training "environment". By penalizing distributional deviation, we shift the response distribution closer to where the reward model is most reliable without modifying the base policy itself. This means that as long as the responses used to train the pessimism network are representative of where the reward model works well, we can expect improvements when applying it to other policies, since we're effectively doing this "soft" distributional guidance. The key insight is that the predictor learns what "typical" responses look like from the reward model's perspective, not from any specific policy's perspective, so it can generalize across different base models as long as they produce responses within the reward model's domain of reliability.

---

### Meta-Review · Area_Chair_uNkE · 2026-01-05

**Summary:**

This meta-review recommends the acceptance of the paper. The consensus among the reviewers is that the work addresses a significant and timely problem: the degradation of performance in Best-of-N (BoN) sampling due to reward hacking. The proposed "caution" mechanism—a lightweight, inference-time technique that penalizes out-of-distribution (OOD) responses—is praised for its elegance, simplicity, and practical efficiency. Reviewers were particularly pleased by the intuitive framing of the method and the strength of the empirical results across reasoning benchmarks like GSM8K and MATH-500. While early concerns were raised regarding the theoretical justification and the lack of comparison with existing uncertainty-aware baselines, the general sentiment is that the paper provides a robust and well-validated solution to a persistent challenge in LLM alignment.

**Reviewer Concerns:**

The authors were highly proactive during the rebuttal period, successfully addressing the majority of the committee’s concerns. Reviewer’s request for code and baselines was met with an anonymous repository and a comparison against $\chi^2$-regularization and soft BoN, leading to a score increase. Similarly, reviewers' concerns regarding notation and the "monotonicity" of performance gains were resolved through manuscript revisions. A remaining point of contention involves Reviewer's request for a comparison with self-consistency. However, the reviewers largely agrees with the authors’ reasoning that self-consistency is orthogonal to the study of reward-guided selection, as it does not utilize reward models and thus cannot "hack" them. While the ablation studies on layer sensitivity were performed on subsets due to computational constraints, the emerging trends were deemed sufficiently clear to support the paper’s claims.

**Reviewer Scores:**

Reviewers 7MEf and JSWD are already firm in their support, having had their technical questions regarding notation and systematic bias answered. Reviewer sWFV moved from a 4 to a 6 following the code release and is likely satisfied. Reviewer 4Xb3 did not respond to the rebuttal, but given that the authors provided the specific $\chi^2$ and soft BoN baselines they requested, it is highly probable they would have increased their score to a 7 or 8. Reviewer nUU2 remains the lone outlier; however, their primary critique regarding self-consistency is viewed by the rest of the committee as a misunderstanding of the "reward hacking" research scope.Overall, the technical contribution and empirical depth of the work merit its inclusion in the program.

---

### Decision · Program_Chairs · 2026-01-26

Accept (Poster)